# Human-Induced Pluripotent Stem Cell-Derived Neural Progenitor Cells Showed Neuronal Differentiation, Neurite Extension, and Formation of Synaptic Structures in Rodent Ischemic Stroke Brains

**DOI:** 10.3390/cells13080671

**Published:** 2024-04-12

**Authors:** Yonehiro Kanemura, Atsuyo Yamamoto, Asako Katsuma, Hayato Fukusumi, Tomoko Shofuda, Daisuke Kanematsu, Yukako Handa, Miho Sumida, Ema Yoshioka, Yutaka Mine, Ryo Yamaguchi, Masayasu Okada, Michihiro Igarashi, Yuko Sekino, Tomoaki Shirao, Masaya Nakamura, Hideyuki Okano

**Affiliations:** 1Department of Biomedical Research and Innovation, Institute for Clinical Research, NHO Osaka National Hospital, Osaka 540-0006, Japan; yamamoto.atsuyo.qy@mail.hosp.go.jp (A.Y.); katsuma.asako.gh@mail.hosp.go.jp (A.K.); fukusumi.hayato.zt@mail.hosp.go.jp (H.F.); sumida.miho.yc@mail.hosp.go.jp (M.S.);; 2Department of Neurosurgery, NHO Osaka National Hospital, Osaka 540-0006, Japan; 3Department of Neurosurgery, NHO Tokyo Medical Center, Tokyo 152-8902, Japan; ymine@keio.jp; 4Department of Physiology, Keio University School of Medicine, Tokyo 160-8582, Japan; ryo.yamaguchi@sumitomo-pharma.co.jp (R.Y.); hidokano@a2.keio.jp (H.O.); 5Regenerative & Cellular Medicine Kobe Center, Sumitomo Pharma Co., Ltd., Kobe 650-0047, Japan; 6Department of Brain Tumor Biology, Brain Research Institute, Niigata University, Niigata 951-8585, Japan; masayasu_okd@bri.niigata-u.ac.jp; 7Department of Neurosurgery, Brain Research Institute, Niigata University, Niigata 951-8585, Japan; 8Department of Neurochemistry and Molecular Cell Biology, School of Medicine, Graduate School of Medical, Dental Sciences Niigata University, Niigata 951-8510, Japan; tarokaja@med.niigata-u.ac.jp; 9Department of Veterinary Pathophysiology and Animal Health, Graduate School of Agricultural and Life Sciences, The University of Tokyo, Tokyo 113-8657, Japan; yukos@g.ecc.u-tokyo.ac.jp; 10AlzMed, Inc., Tokyo 113-8485, Japan; tshirao@alzmed.jp; 11Department of Orthopaedic Surgery, Keio University School of Medicine, Tokyo 160-8582, Japan; masa@a8.keio.jp; 12Keio Regenerative Medicine Research Center, Keio University, Kawasaki 210-0821, Japan

**Keywords:** ischemic stroke, induced pluripotent stem cells, neural progenitor cells, intracerebral transplantation, neuronal differentiation, neurite extension, synaptic structure formation, cell replacement effect

## Abstract

Ischemic stroke is a major cerebrovascular disease with high morbidity and mortality rates; however, effective treatments for ischemic stroke-related neurological dysfunction have yet to be developed. In this study, we generated neural progenitor cells from human leukocyte antigen major loci gene-homozygous-induced pluripotent stem cells (hiPSC-NPCs) and evaluated their therapeutic effects against ischemic stroke. hiPSC-NPCs were intracerebrally transplanted into rat ischemic brains produced by transient middle cerebral artery occlusion at either the subacute or acute stage, and their in vivo survival, differentiation, and efficacy for functional improvement in neurological dysfunction were evaluated. hiPSC-NPCs were histologically identified in host brain tissues and showed neuronal differentiation into vGLUT-positive glutamatergic neurons, extended neurites into both the ipsilateral infarct and contralateral healthy hemispheres, and synaptic structures formed 12 weeks after both acute and subacute stage transplantation. They also improved neurological function when transplanted at the subacute stage with γ-secretase inhibitor pretreatment. However, their effects were modest and not significant and showed a possible risk of cells remaining in their undifferentiated and immature status in acute-stage transplantation. These results suggest that hiPSC-NPCs show cell replacement effects in ischemic stroke-damaged neural tissues, but their efficacy is insufficient for neurological functional improvement after acute or subacute transplantation. Further optimization of cell preparation methods and the timing of transplantation is required to balance the efficacy and safety of hiPSC-NPC transplantation.

## 1. Introduction

Ischemic stroke is a major cerebrovascular disease with a high morbidity and mortality rate that results in various neurofunctional disabilities, including motor, sensory, language, and cognitive functions. Ischemic stroke develops because of several factors, including blood clot embolisms formed by irregular heartbeats such as atrial fibrillation, thrombosis by a thrombus developed at the fatty plaque within the atherosclerotic blood vessel, or cerebrovascular malformations. Recent progress in early recanalization therapies for acute ischemic stroke (AIS) using intravenous thrombolysis with a recombinant tissue plasminogen activator or intra-arterial catheter-based mechanical endovascular thrombectomy has significantly reduced mortality and improved prognosis in selected patients [1,2,3].

However, brain tissue ischemia can cause irreversible damage to neural tissues, leading to persistent neurological dysfunction and severely affecting the patient’s quality of life. Effective treatments to restore damaged neurological functions are yet to be developed, and novel therapeutic strategies are strongly desired.

A novel and promising strategy to restore a damaged central nervous system (CNS) is regenerative medicine using cell-based therapy [4,5,6,7,8,9]. To date, different types of cells or tissues have been transplanted into ischemic stroke-damaged brains via different routes of access (intravenous, intra-arterial, intrathecal, or direct intracerebral administration) at different stages (acute, subacute, or chronic) and their efficacy in functional recovery has been validated in non-clinical and clinical studies [4,5,6,7,8,9]. Historically, the earliest approach was the intracerebral transplantation of fetal and neonatal brain tissues. A series of non-clinical studies have shown that grafted fetal neural tissues survive, form functional synaptic connections in the host brain, and therefore contribute to the improvement in stroke-induced neurological function [10,11,12]. Due to ethical concerns regarding the use of fetal tissues, human fetal neural tissue transplantation has almost never been used in clinical research for the treatment of ischemic stroke. Human embryonal carcinoma-derived neurons (hNT neurons) [13] and fetal porcine cells [14] have been clinically used as more ethically acceptable cell sources. Although these early clinical trials showed promising findings, they have been discontinued.

In the 2000s, various non-neural lineage cells such as mesenchymal stem cells (MSCs), CD34-positive hematopoietic stem cells (HSCs), umbilical cord blood (UCB), and bone marrow (BM)-derived mononuclear cells (MNCs) were investigated as potential sources of cell-based therapy for ischemic stroke [4,5,6,7,8,9]. Many non-clinical studies have shown that cell therapies using these non-neural lineage cells contribute to significant functional improvement in ischemic stroke-induced neurological dysfunction and the clinical efficacy of these protocols is being actively evaluated in a number of clinical trials [4,5,6,7,8,9]. Several encouraging results have been reported; however, these results are still inconsistent and controversial, and further continuous evaluation is needed [4,5,6,7,8,9]. In addition, neural stem and/or progenitor cells (NSPCs) isolated from neural tissues [15,16,17,18], neural progenitor cells (NPCs) differentiated from embryonic stem cells (ESCs) [19,20], and induced pluripotent stem cells (iPSCs) [21,22,23,24,25,26,27,28,29,30,31,32], have also been investigated as cellular sources for cell-based therapies for ischemic stroke. Intracerebral transplantation of human fetal brain tissue-derived NSPCs (hf-NSPCs) has shown promising effectiveness in regard to improving neurological functions damaged by ischemic stroke [33]. However, evidence on the use of NSPCs or NPCs is still scarce and not yet fully elucidated, and regenerative medicine for ischemic stroke using ESCs or iPSC-derived cells has hardly been processed to the clinical stage.

Recently, we developed a culture protocol to generate human iPSC-derived NPCs (hiPSC-NPCs) based on a dual SMAD inhibition method [34,35] and evaluated their safety in vivo [36]. We then established a platform to generate clinical-grade hiPSC-NPCs from integration-free hiPSCs from donors who were homozygous for human leukocyte antigen (HLA)-A, HLA-B, and HLA-DR alleles (HLA-homo) [37], and a first-in-human clinical trial of the transplantation of HLA-homo hiPSC-NPCs in subacute complete spinal cord injury was initiated [38]. In this study, we transplanted HLA-homo hiPSC-NPCs, which were prepared using the same culture protocol and quality standards as those used for clinical application [38], in ischemic stroke-damaged brain tissues at both the acute and subacute stages and evaluated their histopathological properties and usefulness for improving neurological dysfunction.

## 2. Materials and Methods

### 2.1. Neural Induction of hiPSCs

HLA-homo (HLA-A*24:02, B*52:01, and DR*15:02) hiPSCs, clone Ff-WJ14s01 (Appendix A), established from umbilical cord blood cells using an integration-free reprogramming method [37,39] and a feeder-free protocol [40] at the Center for iPS Cell Research and Application (CiRA), Kyoto University, were used to generate hiPSC-NPCs. The neural induction of these hiPSCs was performed using a dual SMAD-inhibition method with LDN-193189 (100 nM; REPROCELL, Kanagawa, Japan) plus SB431542 (10 μM; FUJIFILM Wako Pure Chemical Corporation, Osaka, Japan) as previously described [36,38]. These hiPSC-NPCs were propagated using the neurosphere method [41] in DMEM/Ham’s F-12 (DMEM/F12; FUJIFILM Wako Pure Chemical Corporation) supplemented with epidermal growth factor (EGF: 20 ng/mL; R & D Systems, Minneapolis, MN, USA), fibroblast growth factor 2 (FGF2: 20 ng/mL; R & D Systems), leukemia-inhibitory factor (LIF: 10 ng/mL; Creative BioMart, Shirley, NY, USA), gamma ray-irradiated B-27 Supplement, Xenofree (2%; Thermo Fisher Scientific, Waltham, MA, USA), and heparin (1/1000 dilution; Yoshindo, Osaka, Japan) (neurosphere culture medium) [36,38]. After four passages, the expanded HLA-homo hiPSC-NPCs were suspended in a Stem Cell Banker (Takara Bio Inc., Shiga, Japan) and cryopreserved in liquid nitrogen until transplantation. As controls, three hf-NSPCs isolated from fetal forebrain tissue (oh-NSC-2-fb, oh-NSC-3-fb, and oh-NSC-7-fb) [41] and two hiPSC-NPCs (1210B2 and 1201C1) generated by the same differentiation method [36] were used.

### 2.2. Transcript Analyses

Total RNAs were isolated by QIAzol Lysis Reagent (Qiagen, Valencia, CA, USA), and cDNA was synthesized using a PrimeScript^®^ RT reagent Kit (Takara Bio Inc.) according to the manufacturer’s specification. Quantitative PCR analysis was performed using TaqMan Gene Expression Assays (Thermo Fisher Scientific, Appendix A) with TaqMan™ Gene Expression Master Mix (Thermo Fisher Scientific) and QuantStudio™ 12 K Flex Real-time PCR system (Thermo Fisher Scientific). Gene expression was quantified using the comparative Ct method [36,42].

### 2.3. Flow Cytometric (FCM) Analyses

FCM analyses were performed after the preparation of single-cell suspensions using TrypLE Select Enzyme (Thermo Fisher Scientific). For the analysis of intracellular marker expression, cells were fixed and permeabilized using True-Nuclear Transcription Factor Buffer (BioLegend, San Diego, CA, USA). The cells were then incubated with primary antibodies (Appendix A) for 30 min (min) at room temperature (RT) and further incubated with secondary antibodies conjugated to AlexaFluor-488 (Thermo Fisher Scientific) for 30 min as needed. The stained samples were analyzed using a BD FACSVerse (BD Biosciences, San Jose, CA, USA).

For cell cycle analysis, cells were reacted with propidium iodide solution (final concentration: 10 µg/mL, Merck, Darmstadt, Germany) to stain double stranded DNA, and analyzed by an EC800 Analyzer (Sony Biotechnology Inc., Tokyo, Japan) and ModFit LT (version 4.0, Verity Software House, Bedford, MA, USA).

### 2.4. Cell Proliferation Analysis

Cell growth was evaluated using the CellTiter-Glo Luminescent Cell Viability Assay (Promega, Madison, WI, USA), according to the manufacturer’s instructions. Luminescence intensity was measured using an ARVO X5 Multilabel Plate Reader (PerkinElmer, Waltham, MA, USA) on days 0 and 3–7 after cell seeding to monitor cell growth, and doubling time was calculated using an exponential function approximation formula [36,41].

### 2.5. Karyotype Analysis

Karyotype analysis was performed using conventional Giemsa staining and G-banding as previously described [36]. Chromosome counting was performed for 50 cells using conventional Giemsa staining, and the karyotype was determined based on the G-banding results of more than 20 cells [38]. Stained cells were observed under a microscope (Carl Zeiss, Oberkochen, Germany) and analyzed using Metafer (version 3.10.3, MetaSystems, Altlussheim, Germany) and Ikaros (version 5.5, MetaSystems). Clonal abnormal karyotype cells were assessed using the ISCN2020 criteria as follows: (1) more than two cells with the same aberration regarding chromosome gain or structural abnormality or (2) three cells with chromosome loss [38,43].

### 2.6. Copy Number Analysis

Copy number aberrations (CNAs) were analyzed using CytoScan HD Array (Thermo Fisher Scientific) [36,38]. Genomic DNAs were extracted using NucleoSpin^®^ Tissue Kit (Machrey-Nagel, Düren, Germany) and then processed according to the manufacturer’s instructions. Scanned data were analyzed using the Chromosome Analysis Suite (Thermo Fisher Scientific) with a high-resolution filter setting, and CNAs in hiPSC-NPCs were compared with their parental hiPSCs [36,38].

### 2.7. In Vitro Differentiation Assay

Neurospheres were dissociated into single cells via incubation with TrypLE Select (Thermo Fisher Scientific) at 37 °C for 5 min and seeded on a Matrigel (Corning, Corning, NY, USA)-coated plate at a density of 7.5 × 10^4^ cells/cm^2^ in DMEM/F12 supplemented with 2% B-27 Supplement (Thermo Fisher Scientific) plus 1% fetal bovine serum (FBS; GE healthcare, Chicago, IL, USA) (the 1% serum medium), or Neurobasal Medium (Thermo Fisher Scientific) supplemented with 2% B-27 Supplement plus 1% GlutaMAX (Thermo Fisher Scientific) (the serum-free medium).

### 2.8. Immunocytochemical (ICC) Analysis

Neurospheres were dissociated into single-cell suspensions using TrypLE Select Enzyme (Thermo Fisher Scientific), fixed in 4% paraformaldehyde (PFA; FUJIFILM Wako Pure Chemical Corporation), and attached to 96-well plates. Samples were permeabilized and blocked with blocking buffer containing phosphate-buffered saline (PBS), 0.1% Triton X-100 (Thermo Fisher Scientific), and 10% normal goat serum (Vector Laboratories, Newark, CA, USA). In vitro differentiated cells were also fixed in 4% PFA at 3 days in vitro (DIV), 14 DIV, or 28 DIV, permeabilized and blocked with blocking buffer. Fixed cells were reacted with primary antibodies (Appendix A) in a blocking buffer overnight at 4 °C. After washing, the cells were incubated with secondary antibodies conjugated to AlexaFluor-488 or 568 (Thermo Fisher Scientific) and DAPI (Dojindo, Kumamoto, Japan) for 1 h at RT.

Stained images were acquired and measured using an ArrayScan XTI HCA Reader with Thermo Scientific HCS Studio 6.6.1 (Thermo Fisher Scientific) or a confocal laser scanning microscope (LSM700; Carl Zeiss). SOX1, nestin (NES), and ELAVL3/4-positive cells were quantitatively shown as a percentage, and GFAP staining was evaluated as positive (+) or negative (−).

### 2.9. Animal Welfare

All animal experimental procedures were performed in a contract research organization, the AAALAC International-accredited facilities of the Hamamatsu Pharma Research, Inc. (Shizuoka, Japan). The protocols were reviewed and approved by the Hamamatsu Pharma Research, Inc. Animal Care and Use Committee (approval nos.: HPRIRB-339 and HPRIRB-340) and were performed in accordance with the Guide for the Care and Use of Laboratory Animals [44].

### 2.10. Rodent Middle Cerebral Artery Occlusion Model

Adult male Wistar rats (6–7 weeks; Charles River Laboratories Japan, Inc., Kanagawa, Japan) weighing 300 ± 10 g were housed in standard laboratory cages (23 °C relative temperature, 50% relative humidity and 12 h light/dark cycle); two to four rats were housed in each cage, with free access to food and water.

Rats were anesthetized with isoflurane and subjected to temporary focal cerebral ischemia via intraluminal proximal middle cerebral artery occlusion (MCAO), as described previously, with slight modification [45,46]. Briefly, the left common carotid artery (CCA), external carotid artery (ECA), and internal carotid artery (ICA) were exposed via blunt dissection, followed by ligation of the CCA and ECA. A 4-0 monofilament nylon thread (Alfresa Pharma, Osaka, Japan) with a silicone-coated tip (diameter 0.38–0.58 mm, length 19 mm) was advanced from the CCA stump via the ICA and up to the origin of the left middle cerebral artery (MCA). The thread was secured in place, the incision was closed, and the anesthesia was stopped. After 120 min of occlusion, the rats were re-anesthetized, and the thread was carefully removed to allow reperfusion. During surgery, the rats were maintained at a body temperature of 37 ± 1 °C by lying on a heating pad. After surgery, all of the animals were orally administered 15 mg/kg cephalexin (FUJIFILM Toyama Chemical Co., Ltd., Tokyo, Japan) for three days.

### 2.11. Intracerebral Transplantation of hiPSC-NPCs

Cryopreserved hiPSC-NPCs were thawed and re-cultured in a neurosphere culture medium for three days. The cells were then cultured for an additional 24 h before transplantation with either γ-secretase inhibitor (GSI), N-[N-(3,5-difluorophenacetyl)-l-alanyl]-S-phenylglycine t-butyl ester (DAPT; Abcam, Cambridge, UK), or none [47].

hiPSC-NPCs were transplanted 2 days (acute transplantation) or 7 days (subacute transplantation) after MCAO surgery. Just before transplantation (the 2 days or the 7 days after MCAO surgery), stratified randomization was performed using SAS Analytics Pro version 9.3 (SAS Institute Japan, Tokyo, Japan) and EXSUS version 8.0 (CAC Croit Corporation, Tokyo, Japan) based on the results of two tests (neurological deficit score and rotarod test), and then the animals were divided into three groups in both transplantation studies: rats treated with GSI-untreated naïve hiPSC-NPCs transplantation [GSI (−) group], GSI-treated hiPSC-NPCs transplantation [GSI (+) group], and without hiPSC-NPCs transplantation [control group].

Rats were anesthetized by intraperitoneal administration of medetomidine (0.15 mg/kg, Meiji Animal Health Co., Kumamoto, Japan), midazolam (2 mg/kg, Sandoz, K.K., Tokyo, Japan), and butorphanol (2.5 mg/kg, Meiji Animal Health Co.). A total volume of 8 μL cell suspension (4 × 10^5^ cells) was intracerebrally injected divided into two points of the ipsilateral left hemisphere, using a stereotaxic frame (Narishige Co., Tokyo, Japan); (1) 4 μL (total 2 × 10^5^ cells) into the striatum [−0.40 mm anterior–posterior (AP), +4.00 mm medial–lateral relative to bregma (ML), and −5.50 mm dorsal–ventral from dura (DV)], and (2) another 4 μL (total 2 × 10^5^ cells) into the cortex (−0.40 mm AP, +4.00 mm ML, and −1.75 mm DV). The control animals were treated with the same volume of Dulbecco’s PBS (D-PBS) as the vehicle. All the animals received daily subcutaneous injections of 10 mg/kg cyclosporine A (Novartis, Basel, Switzerland) for 12 weeks.

### 2.12. Neurological Functional Assessment

Neurological function was evaluated using three different methods. All examinations were conducted using a blinded test, stratified randomization, and cell transplantation. Neurological functional assessments were independently performed by separate researchers.

#### 2.12.1. Neurological Deficit Score (NDS)

Neurological dysfunction in the overall observation of balance, motor coordination, and sensorimotor reflexes for each animal was evaluated using a scoring system modified from Bederson et al.’s study as previously described [48,49]. Forelimb flexion, hindlimb flexion, rotational behavior, lateral displacement, and general posture were observed, and each item was scored as one of 4 degrees (score 0–3). The most severely affected animal scored 15 points (Appendix A). NDSs were evaluated at post-ischemic (post-IS) 1 day, 2 or 7 days (pre-transplantation), and post-transplantation (post-TP) 1, 3, 6, 9, and 12 weeks.

#### 2.12.2. Step Test

Each rat was held firmly by the experimenter, with one hand fixing the rat’s torso and both hind limbs raised 5 cm from the floor. The other hand fixed the forelimb, which was not about to be examined. The rat was stepping using the ischemic side (right) forepaw on a smooth-surfaced table with a width of 90 cm for 20 s in the forehand direction and then moved for 20 s in the backhand direction. Next, the same forehand and backhand direction movements were tested using the healthy side (left) forelimb. The number of steps performed on each forelimb was recorded [46,50]. Step tests were performed pre-ischemic (pre-IS), post-IS 2 or 7 days (pre-transplantation), and post-TP 1, 3, 6, 9, and 12-weeks.

#### 2.12.3. Rotarod Test

The rats were placed on a rotarod device rotating at a speed of 10 rpm, and the time at which the animals completed the trial or fell off the device was recorded (cutoff: 180 s). Two days before the MCAO surgery, all animals were pre-trained, and the animals that could completely walk for 180 s were selected for later studies [46]. Rotarod tests were performed pre-IS, post-IS 2 or 7 days (pre-transplantation), and post-TP 1, 3, 6, 9, and 12 weeks.

### 2.13. Histopathological Analysis

After the final neurological functional assessment at 12 weeks after transplantation, all animals were deeply anesthetized and directly perfused with 10% formalin. Removed brain samples were further fixed in 10% formalin, cut into six coronal sections at 3 mm intervals, and embedded in paraffin. Among them, three sections, which showed significant ischemic changes morphologically, were sliced into 5 μm thickness sections.

Histopathological analyses were performed using hematoxylin and eosin (H&E) staining, Klüver–Barrera (KB) staining, and immunohistochemistry. H&E and KB staining were performed according to standard protocols.

Enzymatic immunostaining was performed using a Leica Bond-Max automatic immunostainer (Leica Biosystems, Nussloch, Germany). Paraffin sections were dewaxed in Bond Dewax solution and rehydrated in alcohol and Bond Wash solution (Leica Biosystems). Antigen retrieval was performed using a 10 mM citrate buffer and pH 6 (ER1) retrieval solution, followed by endogenous peroxidase blocking. After incubation with primary antibodies (Appendix A), detection was performed using the Bond Polymer Refine Detection (DS9800, Leica Biosystems). The sections were counterstained with hematoxylin. The samples were examined using a NanoZoomer-XR C12000 (Hamamatsu Photonics, Shizuoka, Japan).

Immunofluorescence staining was performed using primary antibodies (Appendix A) for 1 h at RT or overnight at 4 °C, and secondary antibodies conjugated with AlexaFluor-488, 555, 568, or 647 (Thermo Fisher Scientific) for 1 h at RT. The samples were examined using a LSM700 or an array scan XTI HCA Reader and Nanozoomer S60 (Hamamatsu Photonics).

### 2.14. Imaging Analysis

The residual brain areas were measured from the images of H&E, KB, and NeuN immunostaining using NDP.view2 (Hamamatsu Photonics), and the infarct areas and ratios were calculated (Appendix A). The thickness of the residual brain tissue was measured at three points on the KB staining images (Appendix A). Immunopositive signals obtained from human nuclear antigen (HNA), Ki-67, and NeuN staining were extracted using NDP.view2 and Adobe Photoshop CS2 (Adobe Inc., San Jose, CA, USA) using ImageJ software (version 1.53a) [51]. Positive pixel count analyses of STEM121 and human synaptophysin (hSYP) were performed using Aperio ImageScope (version 12.4.3.5008, Leica Microsystems, Wetzlar, Germany). Analysis of vesicular glutamate transporter 1 (vGLUT1)-or 2 (vGLUT2)-stained samples was performed using ZEN (version 2012 SP1, Carl Zeiss), and the number of vGLUT1/2-positive pixels in each image was measured using ImageJ (version 1.53a).

Infarct areas and ratios, the thickness of residual brain tissue, and number of NeuN-positive cells were evaluated using three coronal sections as follows: the section centering on the point of transplant (target), +3 mm rostral sections from the target (+3 mm rostral), and +3 mm caudal sections from the target (+3 mm caudal).

### 2.15. Statistical Analysis

Statistical analyses were performed using R-software (version 4.1.1) [52]. Significant differences between the two independent groups were analyzed using the Mann–Whitney U test, and the Steel–Dwass test was used for multiple comparisons. Results of neurological functional assessments were calculated as a relative ratio to baseline results (neurological deficits score: post-IS 1 day, Step test, and rotarod test: 2 or 7 days post-IS) and analyzed using two-way repeated measures analysis of variance (ANOVA). Differences were considered significant if the *p* value was < 0.05.

## 3. Results

### 3.1. Cellular Properties of hiPSC-NPCs Generated from HLA-Homo hiPSCs

We generated hiPSC-NPCs from HLA-homo hiPSCs (HLA-homo hiPSC-NPCs) by combining neural induction with dual SMAD inhibition and neurosphere culture [36,38]. HLA-homo hiPSC-NPCs showed typical neurosphere morphology (Figure 1A), expanded via four passages, and were cryopreserved at the fifth passage. Gene expression of *SOX1* and *PAX*6 was significantly higher, and *POU5F1* (OCT3/4) was expressed at lower levels in HLA-homo hiPSC-NPCs than those in iPSCs (Figure 1B). Other NSPCs-marker genes, *SOX2* and *NES*, tended to be expressed at higher levels than iPSCs, similar to the control NPCs and hf-NSPCs, but the difference was not found to be significant (Figure 1B). ICC analysis revealed most cells expressed both SOX1 and NES expression at the protein level (Figure 1C), and FCM analysis also showed over 90% of cells expressed NSPCs makers as like polysialylated–neural cell adhesion molecule (PSA-NCAM) and GD2 on the cell surface and intracellular SOX1 expression, respectively (Figure 1D). In contrast, the expression of iPSC markers (OCT3/4, TRA1-60, and SSEA3) was sufficiently repressed at both the transcript and protein levels (Figure 1B,D).

HLA-homo hiPSC-NPCs showed good and stable proliferation, and their population doubling time at the point of passage 4 and 5 were 60.3 h and 59.4 h, respectively (Appendix A). Cell cycle analysis showed the percentages of cells in G0/G1, S, and G2/M phases at passage 5 were 65.6%, 34.2%, and 0.3%, respectively (Appendix A), and these results were almost equal and stable between the 4th passages and 5th passages. HLA-homo hiPSC-NPCs had a normal karyotype (Figure 1E), which was confirmed by microarray analysis also showed normal karyotype, and no de novo CNA was generated during the neural differentiation process (Appendix A).

The in vitro differentiation assay showed that HLA-homo hiPSC-NPCs were differentiated into βIII tubulin-expressing neurons or GFAP-expressing glial cells, and quantitatively, more than 60% of cells differentiated into ELAVL3/4 expressing neurons, and differentiation into neurons were more promoted in serum-free condition as the culture time progresses compared with serum contained culture condition (Figure 1F). In addition, some cells differentiated into GFAP-expressing cells, suggesting in vitro gliogenesis from HLA-homo hiPSC-NPCs, and this gliogenesis property was more pronounced in FBS-supplemented culture conditions (Figure 1F).

These findings suggest that HLA-homo hiPSC-NPCs acquired phenotypes similar to NPCs while losing iPSC characteristics, and their proliferative properties were almost stable for four passages without acceleration of proliferation and chromosomal/genomic alterations.

### 3.2. Evaluations of Infarction Size and Numbers of Residual Neurons in Host Brains after Transplantation of HLA-Homo hiPSC-NPCs

To validate the modes of action of HLA-homo hiPSC-NPC transplantation in ischemic stroke, we first neuropathologically evaluated the infarction size and the number of residual host neurons in three sections (target, +3 mm rostral, and +3 mm caudal) of host brains at 12 weeks after transplantation.

In the subacute transplantation, target section-infarct ratios (%) determined from H&E staining images of GSI (−) group, GSI (+) group, and control group were 49.9 ± 6.61, 43.6 ± 5.64, and 50.9 ± 6.71 (mean ± SEM), respectively, and there was no significant difference among them (Figure 2). In the acute transplantation, those of the GSI (−) group, GSI (+) group, and control group were 42.9 ± 7.01, 43.8 ± 7.43, and 53.2 ± 8.39 (mean ± SEM), respectively, and they were also not different significantly (Figure 2). In the other two sections (+3 mm rostral and +3 mm caudal), no significant differences in the infarct ratios were observed in either transplantation study (Figure 2). Similarly, there were no significant differences in the infarct ratios of all three sections, as calculated by KB staining or NeuN immunostaining, in both transplantation studies (Appendix A). Next, we examined the thickness of the residual brain tissues using KB staining images and confirmed that there was no significant difference in the residual brain thickness in all three sections of both transplantation studies (Appendix A). These findings indicate that transplantation of HLA-homo hiPSC-NPCs did not affect the size of the ischemic stroke in subacute or acute transplantation.

To evaluate the number of residual neurons in the host brain tissue, we performed NeuN immunostaining on three sections, calculated the number of immunopositive nuclei, and determined the number of residual neurons by summing the results of the three sections. In the ipsilateral infarct hemispheres, there was no significant difference among the three subacute or acute transplantation groups (Figure 3). In contrast, in both transplantations, the number of residual neurons in the GSI (−) group in the contralateral healthy hemisphere was significantly higher than that in the control group (Figure 3). These findings suggest that transplanted HLA-homo hiPSC-NPCs did not significantly affect the survival of neurons in infarct hemispheres; however, GSI-untreated naïve HLA-homo hiPSC-NPCs might show significant neuroprotective effects in contralateral healthy hemispheres in both transplantations.

### 3.3. In Vivo Survival of the Transplanted HLA-Homo hiPSC-NPCs in the Ischemic Brain

HLA-homo hiPSC-NPCs detected by HNA immunostaining were observed in the host brain tissues 12 weeks after subacute transplantation of both GSI-untreated and GSI-treated cells. There was a trend that the number of HNA-positive (HNA^+^) cells was a few higher in the GSI (+) group; however, there was no statistically significant difference between the two groups (Figure 4A). The number of Ki-67-positive (Ki-67^+^) proliferating cells was low in both groups and was not significantly different (Figure 4A), and the ratio of Ki-67^+^/HNA^+^ cells was not significantly different between the two groups (Appendix A). In acute transplantation, there was no significant difference in the number of HNA^+^ cells between the two groups; however, there was a trend toward higher numbers in the GSI (+) group (Figure 4B). The number of Ki-67^+^ cells after acute transplantation of GSI-treated cells was significantly higher than that of the untreated cells (Figure 4B), although there was no significant difference in the ratio of Ki-67^+^/HNA^+^ cells between the two groups (Appendix A).

These findings indicate that HLA-homo hiPSC-NPCs transplanted at the subacute and acute stages survived in host brain tissues 12 weeks after transplantation, regardless of the presence or absence of GSI treatment; however, their survival rates differed depending on the time of transplantation, which showed that transplantation of GSI-treated cells at the acute stage resulted in higher numbers of Ki-67^+^ cells.

### 3.4. In Vivo Differentiation of the Transplanted HLA-Homo hiPSC-NPCs in the Ischemic Brain

Twelve weeks after subacute transplantation, SOX1-positive (SOX1^+^) cells were detected, but their numbers were small in both the GSI (−) and GSI (+) groups without a statistical difference (Figure 5A). Most HLA-homo hiPSC-NPCs expressed ELAVL3/4 in their nuclei, and the proportion of ELAVL3/4-positive (ELAVL3/4^+^) cells was almost the same between the GSI (−) and GSI (+) groups, indicating predominant differentiation into neurons regardless of GSI treatment (Figure 5A). Furthermore, expression of hSYP, a presynaptic marker, was observed in the ipsilateral infarct hemisphere of both of the groups 12 weeks after transplantation (Figure 5A), and expression of hSYP was also detected in the contralateral healthy hemisphere of the GSI (+) group (Appendix A). The total expression levels of hSYP in the ipsilateral infarct and contralateral healthy hemispheres were almost the same (Figure 5A).

These findings suggest that HLA-homo hiPSC-NPCs transplanted into subacute ischemic brains mainly differentiated into neurons and matured toward synaptic formation.

After the acute transplantation, SOX1^+^ cells were significantly more observed in the GSI (+) group than in the GSI (−) group. Fewer ELAVL3/4^+^ cells were observed in the GSI-treated HLA-homo hiPSC-NPCs, although there was no significant difference compared to the GSI (−) group (Figure 5B). Expression of hSYP in both ipsilateral infarct and contralateral healthy hemispheres was enhanced by GSI treatment of the transplanted cells but not significant (Figure 5B).

These findings suggest that engraftment of HLA-homo hiPSC-NPCs was confirmed 12 weeks after transplantation, but their efficiency and differentiation status differed depending on the timing of transplantation and GSI treatment. Cells transplanted after GSI treatment in the acute stage might show more neuronal maturation, while the efficiency of neuronal differentiation was not higher; rather, significantly higher numbers of proliferating stem cells remained.

### 3.5. Neurites Growth from the Transplanted HLA-Homo hiPSC-NPCs

Twelve weeks after the subacute transplantation, numerous STEM121-positive (STEM121^+^) fibers were detected in the ipsilateral infarct hemisphere (Figure 6A). These fibers expressed both HLA-ABC and growth-associated protein of 43-kDa (GAP-43) phosphorylated at the Thr172 site [pGAP43(Thr172)], which is a marker of neuronal growth cones [53]. This suggests that these STEM121^+^ fibers were human neurites derived from engrafted and differentiated HLA-homo hiPSC-NPCs (Figure 6B, Appendix A). Human HLA-ABC- and pGAP43(Thr172)-positive human neurites were also detected in contralateral healthy hemispheres (Figure 6B). Human neurites originating from HLA-homo hiPSC-NPCs were similarly observed in both the ipsilateral infarct and contralateral healthy hemispheres 12 weeks after acute transplantation (Figure 6A,B).

The total number of STEM121^+^ fiber-like structures in both the ipsilateral infarct and contralateral healthy hemispheres was higher in the GSI (+) group in both the subacute and acute transplantation groups (Figure 6C, Appendix A). In the subacute transplantation, there were no significant differences between the GSI (−) and GSI (+) groups, but those observed in the GSI (+) group tended to spread more broadly from peri-infarct area to remote residual normal areas than those in GSI (−) group (Figure 6C). However, the STEM121^+^ structures observed in the ipsilateral infarct hemisphere after acute transplantation tended to remain locally. In the contralateral healthy hemisphere, the GSI (+) group of the acute transplantation showed significantly higher STEM121^+^ areas than those of the GSI (−) group (Figure 6C).

These findings indicate that HLA-homo hiPSC-NPCs transplanted into ischemic brain tissues differentiated into neurons, broadly extending axons into both the ipsilateral infarct and contralateral healthy hemispheres. The timing of transplantation and GSI treatment of transplanted cells may modify the pattern of axonal extension.

### 3.6. In Vivo Maturation of the Transplanted HLA-Homo hiPSC-NPCs in the Ischemic Brain

Some hSYP-expressing axons formed contacts with drebrin A-expressing structures, which are postsynaptic markers [54], regardless of the timing of the transplantation (Figure 7A). Cluster formations of hSYP and drebrin A were mainly detected in the ipsilateral infarct hemisphere, but a few clusters were observed in the contralateral healthy hemisphere (Figure 7A). These findings indicated that human axons derived from transplanted HLA-homo hiPSC-NPCs form synaptic structures with host rat neurons.

Human neurons derived from GSI treatment HLA-homo hiPSC-NPCs also expressed several other neuronal marker molecules. In the subacute transplantation, most neurons expressed vGLUT1 or 2, which are presynaptic markers of glutamatergic excitatory neurons, and these vGLUT (vGLUT1 or vGLUT2) and hSYP-positive (vGLUT^+^/hSYP^+^) portions were identified in both ipsilateral infarct and contralateral healthy hemispheres (Figure 7B). vGLUT1^+^ areas were larger than those of vGLUT2^+^, but the difference was not significant (Appendix A). vGLUT^+^/hSYP^+^ portions were also observed in the acute transplantation group; however, most cells were identified in the peri-infarct lesion and were difficult to detect in the contralateral healthy hemisphere (Figure 7B). Although the number of individuals that could be used for quantitative analysis was small (n = 2), the vGLUT1^+^ areas in the acute transplantation group were significantly larger than the vGLUT2^+^ areas (Appendix A).

Neurons expressing other markers, such as GAD67 (inhibitory neurons) or choline acetyltransferase (ChAT, cholinergic neurons), were also identified in both subacute and acute transplantation; however, their number was very small, and this meant it was difficult to perform quantitative analysis (Appendix A). In addition to neurons, a small number of cells derived from HLA-homo hiPSC-NPCs expressed human-specific GFAP, as detected by the STEM123 antibody, suggesting differentiation into glial lineage cells (astrocytes; Appendix A). No SOX10-positive cells were observed after either subacute or acute transplantation, indicating that differentiation into oligodendrocytes seldom occurred.

Taken together, these findings show that HLA-homo hiPSC-NPCs predominantly differentiated into vGLUT1/2^+^ glutamatergic excitatory neurons and formed synaptic structures with host neurons in subacute or acute transplantation, and differentiation into other neurons or glial lineage cells is rare.

### 3.7. Neurological Functional Assessment after Intracerebral Transplantation of HLA-Homo hiPSC-NPCs

In the control groups of both acute and subacute transplantation, neurological functions evaluated by NDS transiently deteriorated from baseline (post-IS 1 day) to the time of transplantation (post-IS 2 days or 7 days), and then they gradually recovered for 12 weeks post-transplantation, suggesting that spontaneous recovery occurred in our stroke models (Figure 8). In the subacute transplantation group, the NDS of the GSI (+) group showed better improvement trends, and they significantly improved at 6 weeks (*p* = 0.013) and 9 weeks (*p* = 0.012) compared with the GSI (−) group (Figure 8). In comparison with the control group, the GSI (+) group also showed good trends but did not reach statistical significance, although the *p*-value was 0.0799 at post-TP at 9 weeks (Figure 8). The right forelimb (ischemic side) motor function evaluated using the step test also showed better trends than the control group in the subacute transplantation group, although there was no significant difference. There was no significant difference in the left forelimb (healthy side) motor function in any of the three groups (Figure 8). In acute transplantation, there were no significant differences between the three groups in either the NDS or step tests (Figure 8). We also evaluated neuromuscular coordination using the rotarod test in both transplantation groups; however, there were no significant differences among the three groups (Appendix A).

These findings indicated that HLA-homo hiPSC-NPCs contributed to improving neurological functions damaged by ischemic stroke when they were intracerebrally transplanted after GSI treatment at the subacute phase; however, their effects were modest and not in the acute stage of transplantation.

## 4. Discussion

### 4.1. Exploring Cell-Based Therapies in Ischemic Stroke: From Neuroprotection to Cell Replacement

The contribution of cell-based therapies for the treatment of ischemic stroke has been discussed in two representative mechanisms: (one) the cell replacement effect, which replaces dead and/or damaged host neural cells with transplanted donor cells, and (other) the so-called neuroprotective effects or bystander effects by a variety of trophic factors, cytokines, or exosomes derived from donor cells, which is the leading mode of action, inducing neuroprotective, immunomodulatory, proregenerative, or angiogenesis actions [4,5,6,7,8,9].

Recently, the therapeutic effects of different non-neural lineage cells, such as MSCs derived from BM, UCB, and adipose tissue, or MNCs or HSCs derived from BM or UCB, against ischemic stroke-induced neurological damage have been investigated [4,5,6,7,8,9]. These non-neural lineage cells are recognized as being ethically less controversial and have been generally administered via intravascular routes, which are clinically more accessible routes [4,5,6,7,8,9]. Many non-clinical studies have suggested both therapeutic efficacy and safety as potential cellular sources for cell-based therapies for ischemic stroke, and several clinical studies have supported these possibilities [4,5,6,7,8,9]. However, many non-clinical studies have also shown long-term survival of donor cells in host brain tissues was hardly observed, suggesting that the mode of action of these non-neural lineage cells is predominantly neuroprotective effects or bystander effects induced by the various factors secreted by non-neural lineage cells [4,5,6,7,8,9]. These neuroprotective effects or bystander effects of non-neural lineage cells are expected to contribute to the amelioration of ischemic stroke-induced neurological damage, while it is assumed that their regenerative effects may not be sufficient and further consideration should be simultaneously required to compensate for the biological limitations of non-neural lineage cells.

In contrast, neural lineage cells such as fetal neural tissues [10,11,12], hf-NSPCs [15,16,17,18,33], NPCs generated from ESCs [19,20], and iPSCs [21,22,23,24,25,26,27,28,29,30,31,32] are expected to show cell replacement effects in ischemic brain tissues. Among them, iPSC-NPCs are recognized as ethically less controversial and more acceptable cell sources. Several studies have reported in vivo properties of hiPSC-NPCs or more differentiated cells from hiPSC-NPCs after transplantation into ischemic stroke brain tissues. Oki et al. transplanted hiPSC-derived long-term expandable neuroepithelial-like stem cells (lt-NES) intracerebrally into the stroke-damaged rat brain 48 h (acute stage) after distal MCAO (dMCAO) and observed functional recovery at 1 week after transplantation. Transplanted cells stopped proliferating, survived for at least 4 months without forming tumors, and differentiated into morphologically mature neurons of different subtypes that exhibited the electrophysiological properties of mature neurons and received synaptic input from host neurons [21]. Toenero et al. generated cortical progenitor cells from hiPSC-derived lt-NESs and transplanted them intracortically into the stroke-damaged rat cortex 48 h (acute stage) after dMCAO. Two months after transplantation, the cortically fated cells showed less proliferation and more efficient conversion into mature neurons with morphological and immunohistochemical features of a cortical phenotype and higher axonal projection density compared to non-fated cells, which contributed to neurological function improvement, exhibiting the electrophysiological properties of mature neurons with evidence of integration into the host circuitry [22]. Palma-Tortosa et al. also showed that hiPSC-derived cortical neurons transplanted intracortically at 48 h (acute stage) after dMCAO send extensive axonal projections to both hemispheres of rats with ischemic lesions in the cerebral cortex and demonstrated that myelination of graft-derived axons occurs in the corpus callosum, their terminals form excitatory glutamatergic synapses on host cortical neurons, and that host neurons in the contralateral somatosensory cortex receive monosynaptic inputs from grafted neurons 6 months after transplantation. Activity in grafted neurons mediated by transcallosal connections to the contralateral hemisphere is involved in maintaining normal motor function, suggesting functional integration of efferent projections from grafted neurons into the neural circuitry of the stroke-affected brain [23]. Noh et al. transplanted HLA-homo hiPSC-NPCs into rat brain tissue 7 days after MCAO (subacute stage) and showed that they induced significant behavioral improvements and that a high proportion of transplanted cells survived and differentiated into MAP2-positive mature neurons, GABAergic neurons, or DARPP32-positive medium spiny neurons, forming neuronal connections with striatal neurons in the host brain 12 weeks after transplantation [24]. These reports suggest that hiPSC-NPCs or more differentiated cells transplanted at the acute or subacute stages of ischemic stroke produce cell replacement effects in ischemic brain tissues.

### 4.2. Evaluating the Therapeutic Potential of Transplanted hiPSC-NPCs in Ischemic Stroke Recovery: Successes, Limitations, and Future Directions

In this study, we transplanted HLA-homo hiPSC-NPCs prepared using a combination of dual SMAD-inhibition and neurosphere culture protocol into rodent ischemic brain tissues made by the transient MCAO procedure at either subacute (post-IS 7 day) or acute (post-IS 2 day). We then evaluated their efficacy against the functional improvement in neurological dysfunctions caused by ischemic stroke and the mode of action. hiPSC-NPCs that were intracerebrally transplanted into ischemic brain tissues at either the subacute or acute stage were histologically identified in the host brain tissues 12 weeks after transplantation (Figure 4). Many did not proliferate (Figure 4, Appendix A) and predominantly differentiated into ELAVL3/4^+^, vGLUT1/2^+^, and hSYP^+^ glutamatergic excitatory neurons (Figure 5 and Figure 7). They also favorably extended their neurites into not only the local ischemic hemisphere but also the long-distance contralateral hemisphere, and these neurites from the transplanted hiPSC-NPCs formed synaptic structures with the host neurons (Figure 6 and Figure 7). In addition, some cells expressed other neuronal markers (GAD67, ChAT) and GFAP (Appendix A). Similar to previous reports, these findings show that our HLA-homo hiPSC-NPCs transplanted into ischemic brain tissues had the ability to survive, differentiate, and mature under ischemic brain conditions, suggesting that hiPSC-NPCs contribute to the cellular replacement of lost cells under ischemic brain conditions.

However, several issues have been identified. First, we could not show clear neurological function improvement after hiPSC-NPC transplantation (Figure 8, Appendix A), although most previous reports have suggested significant neurological function improvement after the transplantation of hiPSC-NPCs or their differentiated cells [21,22,23,24,25,26,27,28,29,30,31,32]. In this study, we transplanted the same HLA-homo-hiPSC-NPCs using two different methods (GSI-untreated vs. GSI-treated) at two different time points (acute vs. subacute). Among these combinations, transplantation of GSI-pretreatment cells at the subacute stage showed some positive effects on neurological functional assessments, and we believe that transplantation of our HLA-homo hiPSC-NPCs had positive effects on ischemic stroke brain tissues, when evaluated together with histopathological results. It has been reported that the GSI pretreatment of hiPSC-NPCs promotes neuronal differentiation and maturation in vitro, and GSI pretreatment also reduces the overgrowth of transplanted hiPSC-NPCs and inhibits the deterioration of motor function in vivo in the case of transplantation for spinal cord injury [47]. Based on this evidence, the GSI pretreatment of hiPSC-NPCs was used for our ongoing clinical trial for subacute spinal cord injury [38]. The present study indicated the usefulness of GSI-pretreated cells for transplantation at the subacute stage of stroke and that GSI-pretreatment of transplanted cells might be an effective therapeutic procedure for the regenerative treatment of subacute stroke.

However, our findings also suggest that neurological functional assessments in rodent models after acute or subacute stage transplantation may be complicated and difficult to evaluate. Rodent neurological function impairments sometimes spontaneously and greatly recover, even without hiPSC-NPC transplantation, after ischemic stroke (Figure 8, Appendix A). The effects of the cell transplantation may be masked by a larger spontaneous recovery and become obscure. As a result, we could only recognize the effects of cell transplantation as modest and could not find statistical differences compared to controls. It may be recommended that neurological functional assessment of rodent stroke models after cell transplantation at the acute or subacute stage should be performed using a larger number of animals by methods with a higher sensitivity, which needs to be further evaluated in future studies.

Next, the present results suggested the neuroprotective or bystander effects of transplanted hiPSC-NPCs. It has been reported that hiPSC-NPC transplantation results in enhanced endogenous repair processes, including decreases in post-stroke neuroinflammation and glial scar formation and an increase in proliferating endogenous neural stem cells in the subventricular zone as well as the perilesional capillary networks, which suggests the existence of neuroprotective effects or bystander effects induced by transplanted hiPSC-NPCs [24]. In contrast, transplantation of our HLA-homo hiPSC-NPCs displayed no significant effect on the size of ischemic stroke and survival of ipsilateral infarct hemisphere neurons in the subacute or acute transplantation groups (Figure 2 and Figure 3, Appendix A). Interestingly, we found that the number of host neurons in the contralateral healthy hemispheres was significantly higher after transplantation of GSI-untreated naïve HLA-homo hiPSC-NPCs (Figure 3). These findings indicate that our HLA-homo hiPSC-NPCs may exert neuroprotective or bystander effects in both ipsilateral infarcts and contralateral healthy hemispheres when transplanted without GSI pretreatment. However, their effects are not always sufficient to protect against and repair ischemic stroke-induced damage, and the significance of the neuroprotective or bystander effects of transplanted hiPSC-NPCs should be evaluated in future studies.

In addition, the present study highlights the importance of the timing of cell transplantation. In the subacute stage of transplantation, hiPSC-NPCs showed significant cell replacement effects and also some positive effects for improving neurological functions by GSI pretreatment before transplantation (Figure 8, Appendix A). In contrast, in acute-stage transplantation, hiPSC-NPCs showed no positive effects on improving neurological functions with or without GSI pretreatment before transplantation (Figure 8, Appendix A), and higher numbers of Ki-67^+^ proliferating or SOX1^+^ immature cells significantly remained in GSI-treated cell transplantation (Figure 4B and Figure 5B), although neurite growth into the contralateral healthy hemisphere from transplanted cells was significantly enhanced compared to GSI-untreated cell transplantation (Figure 6C). These findings indicate that the merits of hiPSC-NPC transplantation are small and not significant in acute stage transplantation, even with GSI pretreatment before transplantation, but may increase the risk of undifferentiated and immature transplanted cells. These different results might arise from differences in the pathophysiological conditions between the subacute and acute stages of ischemic stroke. The timing of the cell transplantation and preparation methods for the transplanted cells should be further investigated, and optimization is required to balance the efficacy and safety of hiPSC-NPC transplantation.

## 5. Conclusions

hiPSC-NPCs transplanted intracerebrally into rodent transient MCAO ischemic models at either the subacute or acute stage were histologically identified in host brain tissues, showing neuronal differentiation, neurite extension, and the formation of synaptic structures at 12 weeks after transplantation. They had some positive effects on improving neurological functions when transplanted at the subacute stage with GSI pretreatment, but their effects were modest and not significant and showed a possible risk of remaining in their undifferentiated and immature status in acute stage transplantation. These findings suggest that hiPSC-NPCs show cell replacement effects in ischemic stroke-damaged neural tissues, but their efficacy is insufficient for neurological functional improvement in the use of acute or subacute transplantation. Further investigation and optimization of the timing of cell transplantation and the preparation methods of transplanted cells are needed to consider the balance of efficacy and safety of hiPSC-NPC transplantation.

## Figures and Tables

**Figure 1 cells-13-00671-f001:**
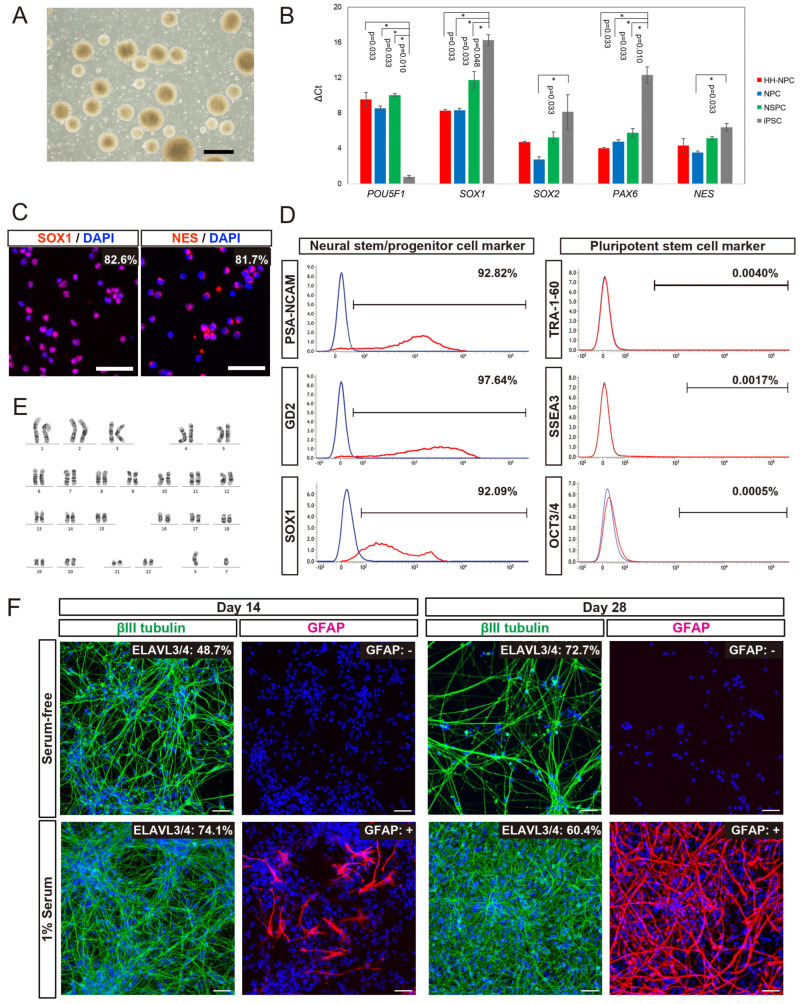
Cellular properties of HLA-homo hiPSC-NPCs. (**A**) Representative phase contrast image. Bar = 500 μm. (**B**) Gene expression analysis using TaqMan assays; HLA-homo hiPSC-NPCs (HH-NPC: biological duplicate, technical duplicate; total n = 4), control hiPSC-NPCs (NPC: biological duplicate, technical duplicate; total n = 4), hf-NSPCs from fetal forebrain tissues (NSPC: biological triplicate, technical duplicate; total n = 6), and undifferentiated iPSCs (iPSC: biological duplicate, technical duplicate; total n = 4). Bar graphs are shown by mean ± SEM. (**C**) Immunocytochemical analysis of SOX1 and nestin (NES). Bar = 50 μm. (**D**) Flow cytometric analyses: red lines are the stained signal, and blue lines are negative controls. (**E**) Karyotype analysis carried out by G-banding. (**F**) In vitro differentiation assay. βIII tubulin (green), GFAP (red). Blue is nuclear staining using DAPI. Percentage of ELAVL3/4-positive cells was quantitatively determined independently. GFAP staining was evaluated by positive (+) or negative (−). Bar = 50 μm.

**Figure 2 cells-13-00671-f002:**
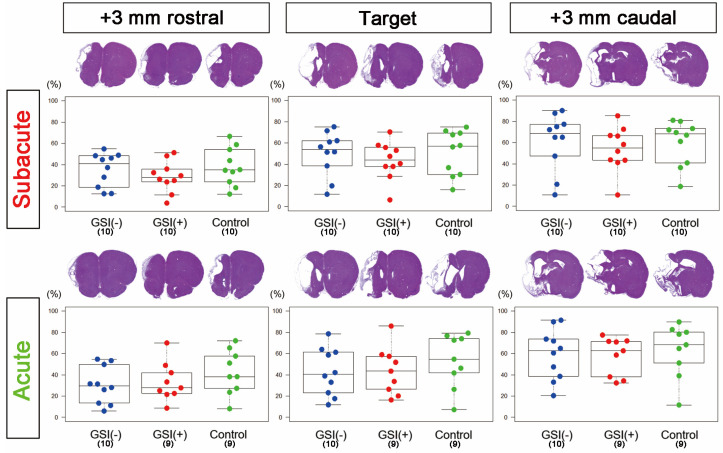
Infarct ratios in subacute and acute transplantation studies. The infarct ratio of each animal was measured using three hematoxylin and eosin-stained coronal sections. Target: the section centering on the point of transplant; +3 mm rostral, +3 mm rostral sections from the target; +3 mm caudal, +3 mm caudal sections from the target. The results are shown as box-and-whisker plots. Subacute: GSI (−) group (blue), GSI (+) group (red), and control group (green). Acute: GSI (−) group (blue), GSI (+) group (red), and control group (green).

**Figure 3 cells-13-00671-f003:**
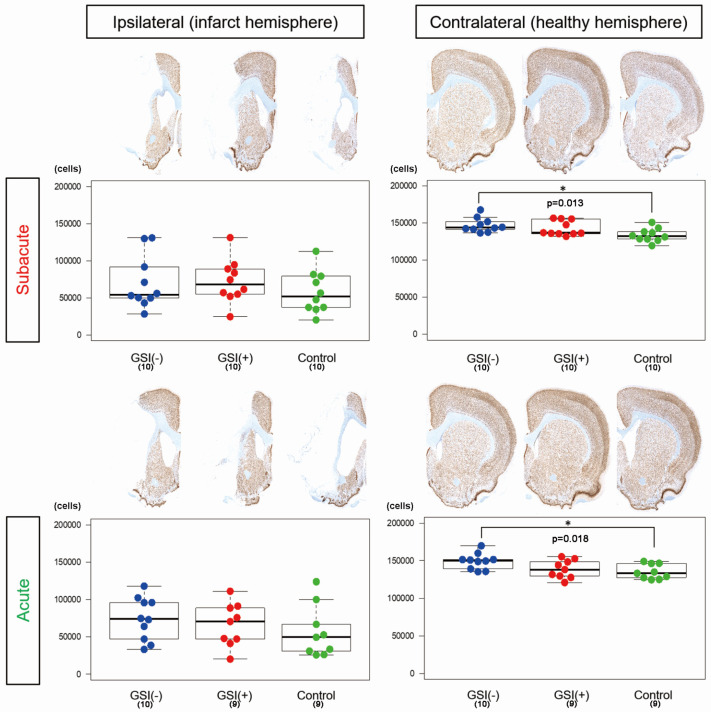
Quantitative assessment of numbers of residual neurons in host brains. The number of residual neurons in the host brain was determined as the sum of NeuN immunopositive nuclei in the three sections. The results are shown as box-and-whisker plots. Subacute: GSI (−) group (blue), GSI (+) group (red), and control group (green). Acute: GSI (−) group (blue), GSI (+) group (red), and control group (green).

**Figure 4 cells-13-00671-f004:**
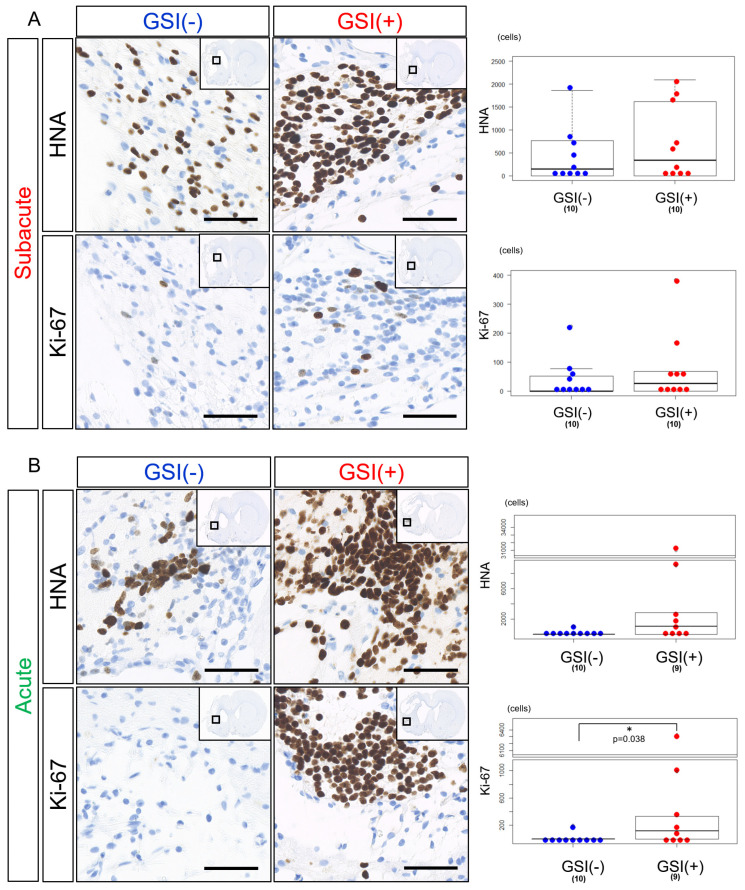
In vivo survival of transplanted HLA-homo hiPSC-NPCs in the ischemic brain. (**A**) HNA- or Ki-67-positive cells in host brain tissues after 12 weeks from subacute transplantation of GSI-untreated or treated HLA-homo hiPSC-NPCs, and the number of positive cells are shown in box-and-whisker plots. GSI (−) group (blue), GSI (+) group (red). Bar = 50 μm. (**B**) HNA- or Ki-67-positive cells in host brain tissues after 12 weeks from acute transplantation of HLA-homo hiPSC-NPCs, and the number of positive cells are shown in box-and-whisker plots. GSI (−) group (blue), GSI (+) group (red). Bar = 50 μm.

**Figure 5 cells-13-00671-f005:**
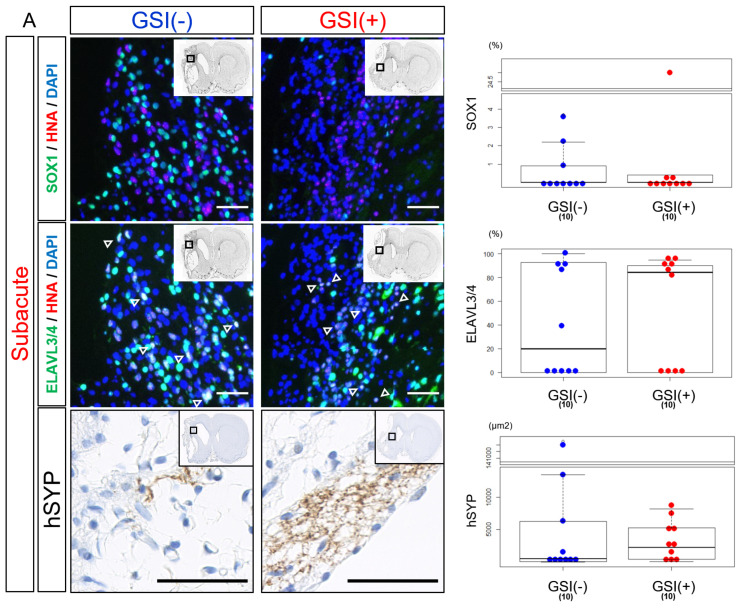
In vivo differentiation of transplanted HLA-homo hiPSC-NPCs in the ischemic brain. (**A**) HNA-positive cells expressing SOX1 or ELAVL3/4 (open triangle) and expression of hSYP in host brain tissues after 12 weeks from subacute transplantation of GSI-untreated or treated HLA-homo hiPSC-NPCs. The results are shown in box-and-whisker plots. GSI (−) group (blue), GSI (+) group (red). Bar = 50 μm. (**B**) HNA-positive cells expressing SOX1 or ELAVL3/4 (open triangle) and expression of hSYP in host brain tissues after 12 weeks from acute transplantation of GSI-untreated or treated HLA-homo hiPSC-NPCs. The results are shown in box-and-whisker plots. GSI (−) group (blue), GSI (+) group (red). Bar = 50 μm.

**Figure 6 cells-13-00671-f006:**
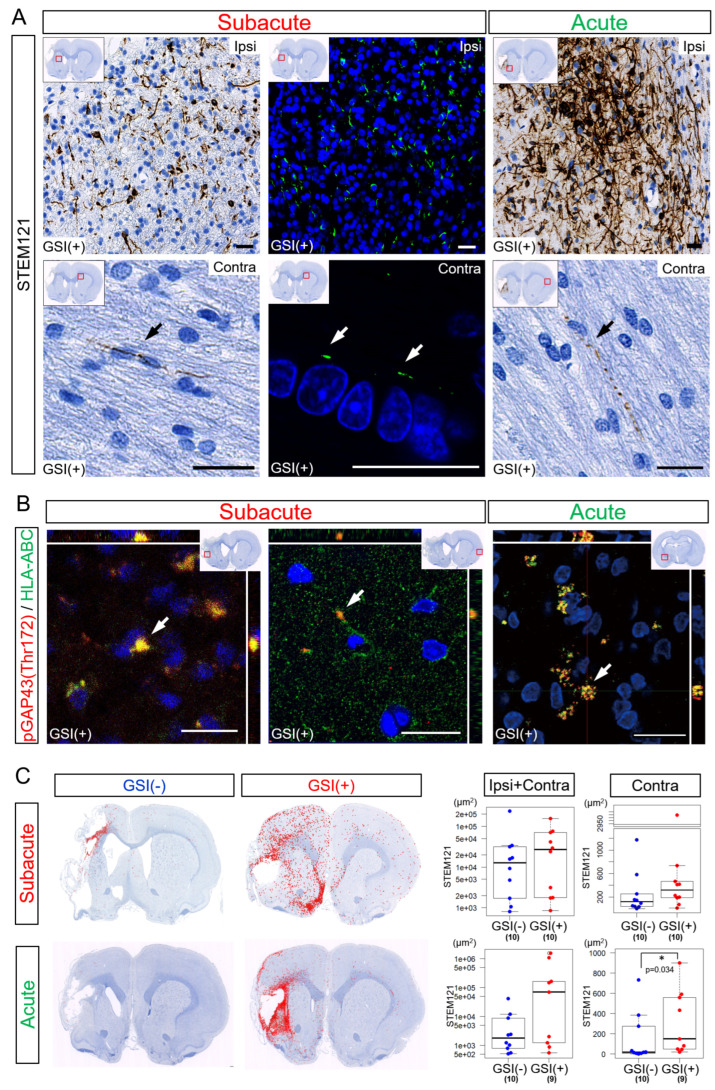
Neurites growth from transplanted HLA-homo hiPSC-NPCs. (**A**) STEM121-positive fibers structures in both the ipsilateral (Ipsi) infarct and contralateral (Contra) healthy hemispheres. Bar = 20 μm. (**B**) Expression of HLA-ABC and growth-associated protein of 43-kDa (GAP-43) phosphorylated at Thr172 site [pGAP43(Thr172)] in both Ipsi and Contra hemispheres. Bar = 20 μm. (**C**) Distribution patterns of STEM121-positive fibers structures. Total amount of STEM121-positive fiber structures calculated by immunopositive pixel count analyses are shown in box-and-whisker plots. GSI (−) group (blue), GSI (+) group (red).

**Figure 7 cells-13-00671-f007:**
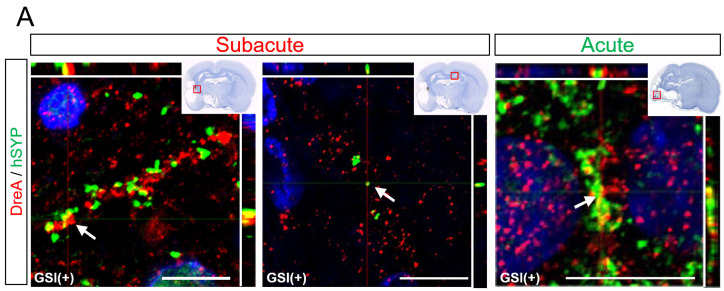
Maturation of the transplanted HLA-homo hiPSC-NPCs. (**A**) Expression of hSYP (green) and synaptic structures (yellow) formed by contact with drebrin A-expressing structures (red). Bar = 10 μm. (**B**) Expression of vGLUT1 or vGLUT2 (red) co-expressed with hSYP (green). Bar = 10 μm. Blue is nuclear staining using DAPI.

**Figure 8 cells-13-00671-f008:**
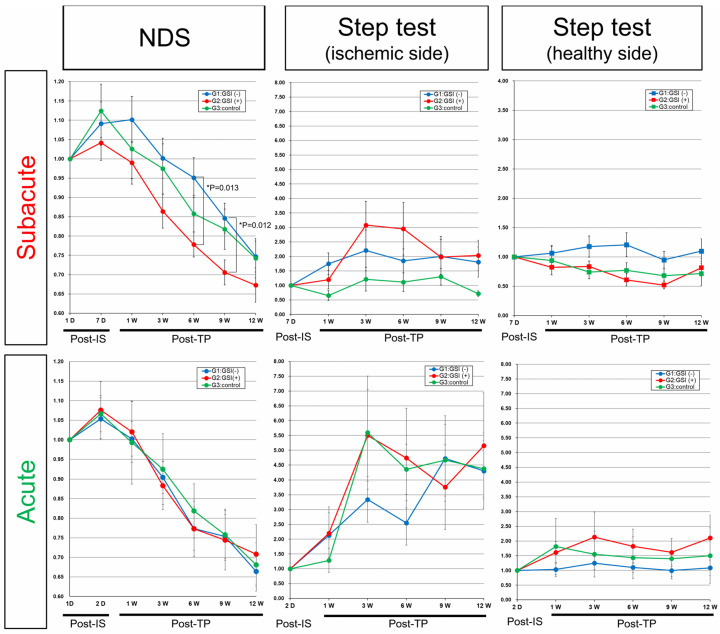
Neurological functional assessment. Results of neurological functional assessments were calculated as a relative ratio to baseline results [Neurological deficit score (NDS): post-IS (ischemic) 1-day; Step test, post-IS 2 days (acute transplantation) or 7 days (subacute transplantation)], and shown by mean ± SEM. Subacute: GSI (−) group (n = 10; blue), GSI (+) group (n = 10; red), and control group (n = 10; green). Acute: GSI (−) group (n = 10; blue), GSI (+) group (n = 9; red), and control group (n = 9; green).

## Data Availability

For any additional questions, please contact the corresponding author.

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
