# Peer review of "Human-Induced Pluripotent Stem Cell-Derived Neural Progenitor Cells Showed Neuronal Differentiation, Neurite Extension, and Formation of Synaptic Structures in Rodent Ischemic Stroke Brains"

_cells, 2024, doi:10.3390/cells13080671_

Round 1

Reviewer 1 Report

Comments and Suggestions for Authors

The manuscript “Human induced pluripotent stem cell-derived neural progenitor cells showed neuronal differentiation, neurite extension, and formation of synaptic structures in rodent ischemic stroke brains” provided objective data and proposed a straightforward viewpoint, regarding the role and action of hiPSC-NPCs in cell-based therapy of ischemic stroke (brain).

Although many studies have reported hiPSC-NPCs transplanted into ischemic animals might improve functional recovery, the current study found that the neurological effects of hiPSC-NPCs were modest and not significant in ischemic brain. Generally, the neurological defects of stroke are directly determined or closely associated with the infarct volume. If stem cell transplantation cannot reduce infarct volume, it necessarily does not improve neurological function. This study demonstrated exactly that relationship.

Considering that differentiation and integration of transplanted cells in host brains is a time-consuming process, it cannot give a rapid compensation for those acute tissue damage resulting from ischemia. Therapeutic values of stem cell transplantation for ischemic stroke or other acute brain damage perhaps is to prevent neurological complications and neurodegeneration, such as cerebral atrophy, depression, dementia, seizures, and Parkinson’s disease. So, further stem cell-based stroke therapy studies may need to evaluate long-term effects of hiPSC-NPCs. Anyway, the current study observed an unusual experimental phenomena which raised the importance of revisiting stem cell therapy for stroke. 

Minor concerns:

Do authors have any data to show cellular properties of hiPSC? In this study, hiPSC are from human samples. Dose the journal need authors to provide Ethical documents? 

Comments on the Quality of English Language

Fine. 

Reviewer 2 Report

Comments and Suggestions for Authors

This study explored the therapeutic effect of human induced pluripotent stem cell-derived neural progenitor cells in a rat model of ischemic stroke. The manuscript was well-written and included detailed information. 

The results showed that stem cells transplanted at the acute or subacute phase were insufficient to improve post-stroke neurological deficits. There was also no significant difference in infarct size between groups.

The adult male Wistar rats (6-7 weeks old) they used had similar body weights, 300 ± 10 g. However, the silicone-coated tips' diameters were 0.38-0.58 mm. Using such a wide range of filament sizes in similar-sized animals will make the lesion vary, requiring a large sample size to detect the intra-group difference.  In addition, the infarct sizes were measured from three slides in each animal. Typically, the entire lesion should be measured, not partially. These factors may alter their conclusions. 
